# Seed Priming with Rhizospheric *Bacillus subtilis*: A Smart Strategy for Reducing Fumonisin Contamination in Pre-Harvest Maize

**DOI:** 10.3390/toxins16080337

**Published:** 2024-07-31

**Authors:** Muhtarima Jannat, Shah Tasdika Auyon, Abu Sina Md. Tushar, Sadia Haque Tonny, Md. Hasibul Hasan, Mangal Shahi, Uday Rana Singha, Ayesha Sultana, Sabera Akter, Md. Rashidul Islam

**Affiliations:** 1Plant Bacteriology & Biotechnology Laboratory, Department of Plant Pathology, Bangladesh Agricultural University, Mymensingh 2202, Bangladesh; jannatppath@gmail.com (M.J.); tushar.1802035@bau.edu.bd (A.S.M.T.); tonny47971@bau.edu.bd (S.H.T.); hasibul.23220507@bau.edu.bd (M.H.H.); mangal.1902405@bau.edu.bd (M.S.); 2Department of Environmental Science, Bangladesh Agricultural University, Mymensingh 2202, Bangladesh; auyon.envsc@bau.edu.bd; 3Department of Agricultural Extension, Khamarbari, Dhaka 1215, Bangladesh; ursingha@dae.gov.bd (U.R.S.); ayesha_sultana07@yahoo.com (A.S.); sabera032@gmail.com (S.A.)

**Keywords:** *Bacillus*, maize, mycotoxin, fumonisin, *Fusarium proliferatum*, biological control, field experiment

## Abstract

Maize, one of the most important cereal crops in Bangladesh, is severely contaminated by fumonisin, a carcinogenic secondary metabolite produced by *Fusarium* including *Fusarium proliferatum*. Biocontrol with *Bacillus* strains is an effective approach to controlling this *F. proliferatum* as *Bacillus* has proven antagonistic properties against this fungus. Therefore, the present study aimed to determine how native *Bacillus* strains can reduce fumonisin in maize cultivated in Bangladesh, where BDISO76MR (*Bacillus subtilis*) strains showed the highest efficacy both in vitro in detached cob and in planta under field conditions. The BDISO76MR strain could reduce the fumonisin concentration in detached cob at 98.52% over untreated control, by inhibiting the conidia germination and spore formation of *F*. *proliferatum* at 61.56% and 77.01%, respectively in vitro. On the other hand, seed treatment with formulated BDISO76MR showed higher efficacy with a reduction of 97.27% fumonisin contamination compared to the in planta cob inoculation (95.45%) over untreated control. This implies that *Bacillus*-based formulation might be a potential approach in mitigating fumonisin contamination in maize to ensure safe food and feed.

## 1. Introduction

Maize or corn (*Zea mays* L.), a crop of the Poaceae family, is an emerging cereal whose popularity has been increasing for diverse uses, i.e., food, fodder, and edible oil [1]. In Bangladesh, maize is the most productive cereal followed by wheat and rice [2]. However, maize cultivation has been severely hampered by *Fusarium* spp., which produces a harmful secondary metabolite known as fumonisins [3]. *Fusarium* spp. that produces fumonisin can infect many crops, foods, and feedstuffs, and its contamination can easily be detectable in the field [4]. The presence of fumonisin in maize grains was found in 15 major maize-producing districts of Bangladesh, where 10 districts exceeded the European Union’s permissible limit for fumonisin [5]. 

The most prevalent fumonisins are FB1, FB2, and FB3, with FB1 being the most poisonous and widely distributed [6]. Animals and humans exposed to FB1 can experience acute toxicity, immunotoxicity, organ toxicity, and reproductive toxicity [7]. Fumonisin exposure has been linked to esophageal and liver cancers in South Africa and Asia; however, the data are inconclusive and it is categorized as possibly carcinogenic to humans [8,9]. Fumonisin is non-genotoxic and disrupts sphingolipid metabolism, affecting various cellular activities. This may contribute to the toxicity and carcinogenicity of fumonisin, as it has structural similarities with the long-chain (sphingoid) base backbones of sphingolipids [6]. Mothers who consumed a lot of fumonisin-contaminated maize during the first trimester of pregnancy had a higher risk of developing neural tube defects in their unborn children [10]. According to studies performed on Tanzanian youth, fumonisin exposure has been linked to stunted growth in children [11,12].

To control the fumonisin contamination in maize, different methods are used such as synthetic fungicides [13], natural resources, plant materials, microbial cultures, genetic materials [3], clay minerals [14], and enzymes [15]. However, it is not easy to find effective and environmentally friendly approaches against the *Fusarium* and mycotoxin production in maize [3]. Researchers are still searching for effective biologically safe alternatives to stop this contaminant from getting into the food chain [3]. However, several rhizobacteria have shown notable antifungal properties to prevent fumonisin contamination in maize plants [16,17]. The *Bacillus* is a rhizobacteria that can survive in various growth conditions and produce aerobically dormant endospores [18]. The different genetic characteristics of *Bacillus* make them anti-pathogenic because they can actively reproduce and withstand adverse conditions [19]. Numerous *Bacillus* strains were found to be efficient biocontrol agents that offered rhizosphere colonization and root disease control, as well as fumonisin contamination reduction in maize grains [20,21,22].

Fumonisin contamination is a significant risk to Bangladesh [23,24,25] and its contamination in maize needs to be monitored to ensure safe food and feed [5]. A recent study showed that maize rhizospheric *Bacillus* can promote plant growth by reducing fumonisin contamination in stored maize grain [26]. Therefore, five *Bacillus* strains (BDISO01RR: *Bacillus amyloliquefaciens*, BDISO76MR: *Bacillus subtilis*, BDISO36PR: *Bacillus subtilis*, BDISO45PR: *Bacillus subtilis* and BDISO49PR: *Bacillus subtilis*) were tested and identified in in vitro conditions, whereas all of them were effective against fumonisin production by colonizing the roots and inhibiting the mycelial growth of *Fusarium proliferatum* on maize grain [27]. So, the present study was carried out to evaluate the potentiality of native *Bacillus* isolates in minimizing fumonisin production under field conditions.

## 2. Results

### 2.1. Effects of Bacillus in Reducing Fumonisin Contamination In Vitro

#### Influence of Different *Bacillus* Isolates on Reducing Fumonisin Accumulation in Co-Inoculated Detached Cobs under In Vitro Condition

An in vitro experiment was conducted to determine how different *Bacillus* isolates can reduce fumonisin concentrations in co-inoculated detached maize cobs. The lowest amount of fumonisin (3.63 ppm) was quantified when maize kernels were treated with BDISO76MR (*B. subtilis*), which was able to reduce the fumonisin concentration by 98.52% over control followed by BDISO01RR (5.06 ppm), BDISO49PR (9.55 ppm), BDISO45PR (33.10 ppm) and BDISO36PR (40.1 ppm) with reduction percentage 97.94, 96.11, 86.52 and 83.67%, respectively as compared to untreated control (Table 1). However, maize kernels treated only with distilled water remained healthy (Figure 1).

### 2.2. In Vitro Ultrastructural Changes in F. proliferatum Hyphae by Bacillus Isolates

At 7 DAI, the PDA plates of *Bacillus* viz. BDISO76MR (*B. subtilis*) and BDISO01RR (*B. amyloliquefaciens*) showed mycelial deformation of *F. proliferatum* among the five isolates. The mycelial structure of *F. proliferatum* was destructed and deformed, including curling, shrinkage, and breakdown by BDISO76MR (*B. subtilis*) and BDISO01RR (*B. amyloliquefaciens*) isolates (Figure 2B,C) as compared to the control. The *F. proliferatum* hyphae in the untreated control were also found to be dense, regular, plump, and undamaged through microscopic examination (Figure 2A).

### 2.3. In Vitro Antagonistic Effect of Bacillus Isolates on F. proliferatum Conidia Germination and Formation

After 12 HPI incubation, the maximum (57.03%) germination reduction in *F. proliferatum* conidia was observed when treated with BDISO76MR followed by BDISO49PR (43.02%), BDISO01RR (34.31%), BDISO36PR (17.34%) and BDISO45PR (3.4%) over untreated control (Table 2 and Figure 3). At 24 HPI, BDISO76MR showed the maximum (61.56%) germination reduction in conidia followed by BDISO01RR (48.33%), BDISO49PR (32.09%), BDISO36PR (22.18%), and BDISO45PR (8.36%) over untreated control. However, the conidia formation at 12 HPI and 24 HPI showed similar reduction rates as at 72 HPI. At 12 HPI, the maximum reduction in conidia formation of *F. proliferatum* was observed in the case of BDISO76MR (78.26%) following BDISO01RR (69.57%), BDISO45PR (60.87%), BDISO49PR (47.83%) and BDISO36PR (30.43%) over untreated control. Similarly, 76.60, 70.21, 61.70, 48.94, and 31.91% reduction in *F. proliferatum* conidia formation was recorded at 24 HPI when treated with BDISO76MR, BDISO01RR, BDISO45PR, BDISO49PR, and BDISO36PR, respectively, over untreated control. At 72 HPI, the maximum (77.01%) reduction in conidial formation of *F. proliferatum* was observed when treated with BDISO76MR followed by BDISO01RR (71.09%), BDISO45PR (60.66%), BDISO49PR (48.82%) and BDISO36PR (31.04%) over untreated control (Table 2). 

### 2.4. Effects of Bacillus in Reducing Fumonisin Contamination in Planta

#### 2.4.1. Fumonisin Concentration of Maize Grains Raised from Treated Seed with *Bacillus* Formulation

To assess the efficacy of *Bacillus*, maize plants were grown from treated seeds with *Bacillus* formulations of five different strains viz. BDISO36PR, BDISO01RR, BDISO49PR, BDISO45PR and BDISO76MR. The minimum (4.50 ppm) fumonisin concentration was quantified in the grains from the plants that treated with BDISO76MR (*B. subtilis*) formulation with 97.27% reduction followed by BDISO01RR (*B. subtilis*) (4.68 ppm) and BDISO36PR (7.77 ppm), BDISO45PR (14.26 ppm), BDISO49PR (33.27 ppm) with reduction rates of 97.16, 95.28, 91.35 and 79.82%, respectively compared to untreated control (Table 3).

#### 2.4.2. Fumonisins Concentration of Maize Grains Obtained from the Co-Inoculated Cob with *F. proliferatum* and *Bacillus* Isolates

The minimum (7.5 ppm) fumonisin concentration was quantified in the grains that were inoculated with BDISO76MR isolate with 95.45% reduction followed by BDISO01RR (9.87 ppm), BDISO49PR (10.26 ppm), BDISO36PR (11.03 ppm), and BDISO45PR (22.94 ppm) with a reduction rate of 94.01, 93.78, 93.31 and 86.09%, respectively as compared to untreated control (Table 3).

## 3. Discussion

*Bacillus*, one of the largest genera of bacteria, is a potential biological control agent because of its antibiotic metabolite production and different genetic characteristics [18,19]. In our laboratory, five *Bacillus* isolates viz. BDISO76MR (*B. subtilis*) from maize rhizosphere, BDISO01RR (*B. amyloliquefaciens*) from rice rhizosphere [28], BDISO36PR (*B. subtilis*), BDISO45PR (*B. subtilis*) and BDISO49PR (*B. subtilis*) from potato rhizosphere [29] were identified and examined their potentiality as a biocontrol agent against *Fusarium proliferatum* of maize in vitro condition [27]. This study revealed the efficacy of these five *Bacillus* isolates to inhibit *F. proliferatum* of maize both in vitro and in planta, where BDISO76MR (*B. subtilis*) showed the highest efficacy in both conditions.

To assess the reduction in fumonisin accumulation, different *Bacillus* strains and *F. proliferatum* were co-inoculated in detached maize cobs in vitro, where BDISO76MR (*B. subtilis*) and BDISO01RR (*B. amyloliquefaciens*) showed the highest reduction in fumonisin accumulation over control by 98.52% and 97.94%, respectively. The reduction in fumonisin concentration might be a result of mycelial growth reduction in *F. proliferatum* by *Bacillus* strains. A similar result was found when combining *F. udum* with *Bacillus* and *Pseudomonas* species resulting in a considerable reduction in mycelial growth and mycotoxin accumulation in vitro [21].

Therefore, an in vitro dual-culture assay was conducted to reveal the effects of *Bacillus* strains on the reduction in radial mycelial growth of the *F. proliferatum,* where BDISO76MR *(B. subtillis)* and BDISO01RR *(B*. *amyloliquefacience*) were found to be the most effective. This reduction in mycelial growth might be because *Bacillus* spp. can produce antimicrobial substances like subtilin, bacilysin, mycobacillin, bacillomycin, mycosubtilin, iturins, fengycins, and surfactins [30]. Although the mechanisms of all the antimicrobial substances are still unrevealed, some researchers have proven how they induce anti-pathogenic properties. Zalila-Kolsi et al. [31] found that the FZB42 strain of *B. amyloliquefaciens* can produce Bacillomycin D, which affects the plasma membrane morphology and cell wall of *F. graminearum* by inducing reactive oxygen species (ROS) accumulation. The inhibitory activity of surfactins was proven against *F. verticillioides*, and *F. oxysporum* because they can produce specific cationic channels in the membrane phospholipid bilayer, which might possess the antibiotic properties [32]. Fengycin, another antimicrobial component produced by *Bacillus* against the filamentous fungi, inhibits the enzyme phospholipase A2 and aromatase functions and it could alter the structure of fungal cell membranes, increasing permeability, and produce permanent lesions that compromise the integrity of fungal cells on *F. graminearum* [33]. However, a different study discovered the inhibitory power of fengycin against different *Fusarium* spp. [34]. On the other hand, iturins demonstrate potent fungi toxic qualities by creating ion-conducting pores when they come into contact with fungal membranes and it is effective against *F. oxysporum* [35,36,37], and *F. graminearum* [31,38]. 

The inhibition of germination and formation of *F. proliferatum* conidia by *Bacillus* strains are an insightful outcome of the experiment. This was observed under a ZEISS Premio Star microscope, whereas the best view of fungal hyphae could be achieved by SEM and TEM analysis [39]. The highest conidiation reduction was observed in BDISO76MR (77.01%), followed by BDISO01RR (71.09%) at 72 HPI over the untreated control. A similar study was conducted in vitro, where *B. subtilis* SG6 exhibited a high antifungal effect on the germination of *F. graminearum* with an inhibition rate of 95.6% over untreated control [40]. Other studies discovered that the *Bacillus* species can induce their antagonistic activity in reducing the mycelial growth of several *Fusarium* species [20], while Baard et al. [41] found *B. subtilis* can suppress mycelial growth of *F. proliferatum* as same as this present study. Gong et al. [38] examined the antagonism of *B. amyloliquefaciens* S76-3 in wheat inoculated with *F. graminearum* and found that iturin A can kill the conidia of *F. graminearum* at minimal inhibitory concentration. Therefore, further research is necessary to find out which substances are responsible for the inhibitory power of BDISO76MR (*B. subtilis*) against *F. proliferatum.*

In addition to the biochemical analysis of the antagonistic substances, Adeniji et al. [42] examined seven *Bacillus* isolates with bio-suppressive effects on *F. graminearum* and discovered that those isolates had important gene clusters encoding biocontrol agents. Similarly, Chen et al. [11] characterized the genome of the *Bacillus velezensis* LM2303 strain, which has a strong biocontrol potential against *F. graminearum*. So, the characterization of BDISO76MR (*B. subtilis*) might add more precise information in controlling *F. proliferatum.*

In another study, fumonisin accumulation was evaluated in planta through seed treatment and co-inoculation of maize cob by different *Bacillus* strains. When seeds were treated with five individual *Bacillus* strains, BDISO76MR (*B. subtilis*) was found to be more effective in the case of reduction in fumonisin accumulation in planta maize cob. Again, similar findings were found when maize cobs were co-inoculated with *Bacillus* strains in planta, where the co-inoculation with BDISO76MR (*B. subtilis*) had the highest efficacy, with a reduction in fumonisin accumulation by 95.45% against *F. proliferatum* infection over untreated control. But seed treatment is better than the co-inoculation of *Bacillus* in planta, which might be because of the volatile metabolite produced by *Bacillus* that can trigger the induced systemic resistance in plants and so activate the plant defense mechanism [43]. The host defense response is induced by the metabolites also [31]. Additionally, *Bacillus* species work against fungal pathogens by generating fungi toxic compounds, competing together with them for nutrients, and resulting in systemic acquired resistance in plants [44,45,46]. Our speculation is it would be interesting to see whether these *Bacillus* isolates can induce the expression of some Fumonisin biosynthesized gene in planta. So, the transcriptomic analysis from RNA seq would be the next step to unravel the complex network related to this induced resistance. It is also revealed that fengycin secreted by *Bacillus* could result in structural deformation of the hyphae of *F. graminearum* and reduce proliferation and mycotoxin production in planta, as well as hyphal permeabilization, and determine ear rot development in maize [40,47].

## 4. Conclusions

BDISO76MR (*Bacillus subtilis*) performed best against fumonisin production both in vitro and in planta among the five *Bacillus* isolates. The BDISO76MR (*B. subtilis*) strains demonstrated a powerful inhibitory effect on reducing the fumonisin concentration in inoculated maize cob and the mycelial growth of *Fusarium proliferatum* in vitro. In field conditions, the BDISO76MR (*B. subtilis*) strains showed the highest efficacy for seed treatment and co-inoculation, whereas seed treatment is better than co-inoculation. Therefore, *Bacillus*-based formulation could be a possible strategy to reduce fumonisin contamination in maize farming.

## 5. Materials and Methods

### 5.1. Assessment of the Ability of Bacillus spp. in Reducing Fumonisin Contamination in Maize In Vitro

#### 5.1.1. Bacteria and Fungi Culture Conditions

*Bacillus* isolates, viz. BDISO01PR (*Bacillus amyloliquefaciens*), BDISO36PR (*Bacillus subtilis*), BDISO45PR (*Bacillus subtilis*), BDISO49PR (*Bacillus subtilis*), and BDISO76MR (*Bacillus subtilis*), were used in this study as prepared by Mita et al. [27]. Then, the bacteria isolates were cultured in Luria Bertani (LB) media at 37 °C [39]. On the other hand, previously identified mycotoxigenic *Fusarium proliferatum* strains were stored in our laboratory [5]. The fungal strains of *F. proliferatum* were cultured in potato dextrose agar (PDA) at 25 °C for 7 days. 

#### 5.1.2. Conidial Suspension Preparation for In Vitro Co-Inoculation of Maize Kernels

A total of 1–2 loops from each bacterial isolate were taken in 20 mL distilled water in a falcon tube and the cell density was fixed at 1 × 10^8^ CFU/mL. Subsequently, in another falcon tube, a 3–5 mm mycelial block of *F. proliferatum* from pure culture was added in 20 mL distilled water and the cell density was set at 1 × 10^6^ spores/mL. The conidial suspensions were filtered through double-layer cheese cloth. The cell density was measured with a hemacytometer. 

#### 5.1.3. In Vitro Co-Inoculation of Maize Kernels (Detached Cob) with *F. proliferatum* and Different *Bacillus* Isolates

An in vitro experiment was conducted to detect the effect of different *Bacillus* isolates on *F. proliferatum* in maize kernels. Healthy maize cobs (cv. Everest hybrid maize seed; Company: Bayer Crop Science Ltd. Dhaka, Bangladesh) were harvested at their maturity stage and brought to the laboratory immediately. Then, the cobs were de-husked and cob surfaces were sterilized with 75% ethanol followed by washing with ddH_2_O. The sterilized kernels were scratched using sterilized toothpicks. The maize kernels were then inoculated with 50 µL (10^8^ CFU/mL) of each *Bacillus* suspension (mentioned above) and 50 μL (10^6^ spores/mL) of *F. proliferatum* conidial suspension (mentioned above). However, maize cobs treated with 25% methanol and 100 µL distilled water were considered as controls. The inoculated maize cobs were incubated at 25 °C and 100% humidity for 7 days with 12 h of daylight per day. The experiment was repeated in three replications.

#### 5.1.4. Conidial Suspension Preparation for In Vitro Mycelial Growth Inhibition of *F. proliferatum* by *Bacillus* Isolates

From the bacterial pure culture, 2–3 loops of each isolate were transferred to 250 mL Erlenmeyer flasks in 100 mL sterile water, and cell densities were adjusted to 1 × 10^8^ CFU/mL with the help of a hemacytometer. Fungal conidia suspension was prepared by inoculating 50 mg of fresh mycelia of *F. proliferatum* in 20 mL CMC medium (1 g NH_4_NO_3_, 1 g KH_2_PO_3_, 0.5 g MgSO_4_·7H_2_O, 1 g yeast extract, 15 g carboxymethyl cellulose, and 1 L ddH_2_O) and cultured at 25 °C, 180 rpm for 4 days [39]. The number of conidia in the suspension was determined by 10^6^ spores/mL using a hemacytometer.

#### 5.1.5. In Vitro Mycelial Growth Inhibition of *F. proliferatum* by *Bacillus* Isolates

A dual-culture assay was performed to observe the morphological alterations of the *F. proliferatum* hyphae by different *Bacillus* isolates. The experiment was conducted three times with three replications. Each *Bacillus* isolate was then streaked on a potato dextrose agar (PDA) plate making a triangular loop with a sterile toothpick. Then, a 5 mm *F. proliferatum* mycelial block was placed at the center of the PDA plates. The plates were incubated at 25 °C for 7 days. PDA plates with *F. proliferatum* mycelial block were used as control. After 7 days the changes in hyphal morphology were observed at the growth junction between the edges of the bacterial growth and the fungal colony using a ZEISS Premio Star microscope, and the hyphal morphological structures were captured by ZEISS AxiocamERc 5s lens. 

#### 5.1.6. The Conidia Formation Assay

To determine the conidia formation rate, 5 mm fresh mycelial block of potential *F. proliferatum* and 500 µL *Bacillus* suspensions (10^8^ CFU/mL) were added to 20 mL CMC media and incubated at 25 °C with 180 rpm for 3 days [39]. Here, distilled water was used as control. The number of conidia formed by *F. proliferatum* was counted by a hemacytometer at 12, 24 and 72 h post-inoculation (HPI) and expressed in percentage over control. The experiment was repeated thrice with three replicates of each experiment. 

#### 5.1.7. The Conidia Germination Assay

The conidia formation assay was performed as described by Yu et al. [39]. Briefly, 1 mL of *F. proliferatum* conidial suspension (10^6^ spores/mL) and 500 (10^8^ CFU/mL) of five different *Bacillus* isolates were inoculated in CMC media and cultured at 25 °C and 180 rpm. The 25% (*v*/*v*) methanol was used as a control. When the spores reached at least half the length of the germ tube, it was considered germinated spores. The germination rates were recorded at 12 and 24 h intervals using a ZEISS Premio Star microscope, where a minimum of 200 conidia were counted for each replication. Each experiment was replicated three times.

### 5.2. Field Efficacy of Different Bacillus Isolates in Reducing Fumonisin Contamination

#### 5.2.1. Crop Husbandry Using Seed Treatment

##### Formulation of *Bacillus* Strains

To test the potential of the seed, all of the *Bacillus* strains were formulated using talc powder as described previously by Islam et al. [48]. Firstly, the bacterial strains from their pure cultures were grown on LB agar media for 24 h. The 2–3 loops (10^8^ CFU/mL) from each bacterial isolate were transferred in LB broth for about 6 h. The bacteria cultures were then centrifuged and resuspended in 200 mL peptone broth with bactopeptone. The broth cultures were then kept in a shaker for 2 h to grow. After that, 5 mL sterile 100% glycerol was added to 200 mL broth cultures of *Bacillus*. 

Subsequently, 500 g talcum powder was mixed with 5 g carboxy methyl cellulose and 7.5 g Calcium carbonate. Here, calcium carbonate was used to adjust the pH at 7 and carboxy methyl cellulose worked as an adjuvant. Then, the powder mixture was autoclaved for two consecutive days at 121 °C under 15 PSI pressure for 30 min. 

Then, 200 mL of each bacterial culture (fortified with 1% peptone and 1% glycerol) was added to 500 g of prepared talcum powder. The formulations were then dried overnight in a laminar flow hood. Finally, the formulations were powdered by hand and packed in plastic bags. The formulated bacterial antagonists could be stored at both room and 4–8 °C temperatures in the refrigerator. 

##### Maize Production with Treated Seeds and Management

Hybrid maize seeds (cv. Everest hybrid maize seed; Company: Bayer Crop Science Ltd., Dhaka, Bangladesh) were treated with the above-mentioned formulated *Bacillus* at 10 g/kg maize seeds. After treatment, half an hour was waited to allow the formulations to adhere to the seed surface. In addition, the shelf-life of the best-formulated *Bacillus* was also tested. Control seeds were treated in 200 mL of distilled water only. The treated seeds were then planted manually in a 315 m^2^ (21 m × 15 m) plot of a farmer’s field at Sutiakhali, Mymensingh, Bangladesh following a randomized complete block design (RCBD). The whole plot was divided into three subplots measuring 27 m^2^ (9 m × 3 m) at every 3 m distance. Then, subplots were divided into 10 rows and each row was divided into 10 hills (row-to-row distance 0.75 m and hill-to-hill distance 0.25 m). A single row was used for a single treatment with each subsequent row containing a control. Here, three subplots were considered as three replicates. Two treated maize seeds were sown on each hill. After two weeks when the plants grew, one single plant was kept in every hill. The plants were nourished with proper fertilizer doses following the Fertilizer Recommendation Guide (2018) provided by BARC [49] and irrigated when needed. The cobs were harvested at their maturity and forceps were used to remove kernels from each cob. Five cobs were maintained in each treatment. Lastly, kernels were kept in separate envelopes and refrigerated at 4 °C until analysis.

#### 5.2.2. Maize Production with Co-Inoculation of Maize Cobs with *F. proliferatum* and Different *Bacillus* Isolates in the Field Condition

Fresh hybrid maize seeds of Everest hybrid maize seed (Company: Bayer Crop Science Ltd., Dhaka, Bangladesh) were planted manually in another 315 m^2^ plot of a farmer’s field at Sutiakhali, Mymensingh, Bangladesh following the previous section. Then, 2.5 mL of *F. proliferatum* spore suspension and 2.5 mL of each *Bacillus* suspension (mentioned above in 5.1.2) were injected through the husk leaves into the side of the primary ears (R2 stage) at 120 DAS. Cobs injected with 5 mL of distilled water were considered as control whereas cobs injected with 5 mL of *F. proliferatum* spore suspension were considered as positive control. Five cobs were maintained in each treatment. The infected cobs were harvested at their maturity and sun-dried for 14 days in their respective plots. However, the irrigation was stopped 25 days before cob harvest to induce the drought stress as the fumonisin production increases in drought. The sun-dried cobs were then de-husked and kernels were removed using forceps. Finally, the kernels were placed in separate envelopes and then refrigerated at 4 °C until analysis. 

### 5.3. Quantification of Total Fumonisin in Inoculated Maize Grain (In Vitro and Field)

#### 5.3.1. Sample Preparation for the Determination of Total Fumonisin

The maize grains were prepared for the determination of total fumonisin following the Green Spring total fumonisin Test Kit (Shenzhen Lvshiyuan Biotechnology Co., Ltd. version 2018-01, Dapeng, China) manufacturer’s instructions. All the samples were tested three times with three replications. The representative samples were crushed with a blender and then sieved. An amount of 1 g of sieved samples from each was then measured in a sanitized conical flask. An amount of 1.0 ± 0.05 g of sieved samples was added to 5 mL of sample extract solution in a 50 mL centrifuge tube, followed by 3 min of agitation, and then centrifuged for 10 min, 20 °C at a speed of over 4000 rpm/min. After that, 700 µL sample redissolving solution was added and finally, 100 µL of the supernatant was collected. However, only 50 µL was used to quantify the total fumonisin.

#### 5.3.2. Total Fumonisin Determination Assay

A volume of 50 µL of the enzyme conjugate solution was added to each well after 50 µL of the sample or the standard solution to separate duplicate wells. Next, 50 µL of the antibody working solution per well was added. The plate was gently mixed by shaking. After being coated with a membrane, the microplates were sealed and left in the dark for 30 min at 25 °C. The cover membrane was then opened carefully, and the liquid was poured out of the microwell. Then, 250 µL/well washing buffer was added and washed fully 4–5 times for 15–30 min. Then, the microwells were taken out and flapped to dry with absorbent paper. Each well received 50 µL of the Substrate A solution and then the Substrate B solution. After gently shaking the plates, they were incubated for 15 min at 25 °C in the dark. Following the addition of 50 µL of the stop solution to each well, the plate was gently shaken to mix. The Green Spring Fumonisin B1 ELISA Test Kit (Shenzhen lvshiyuan Biotechnology Co., Ltd., version 2018-01, Dapeng, China) was used to perform an Enzyme-Linked Immunosorbent Assay (ELISA) (detection limit 2 ppb) using 50 µL of extract. The samples with higher concentrations were diluted when necessary. Then, the wavelengths of the microplates were set at 450 nm and 630 nm and the absorbance was measured using an ELISA reader (model: HEALES MB-580).

## Figures and Tables

**Figure 1 toxins-16-00337-f001:**
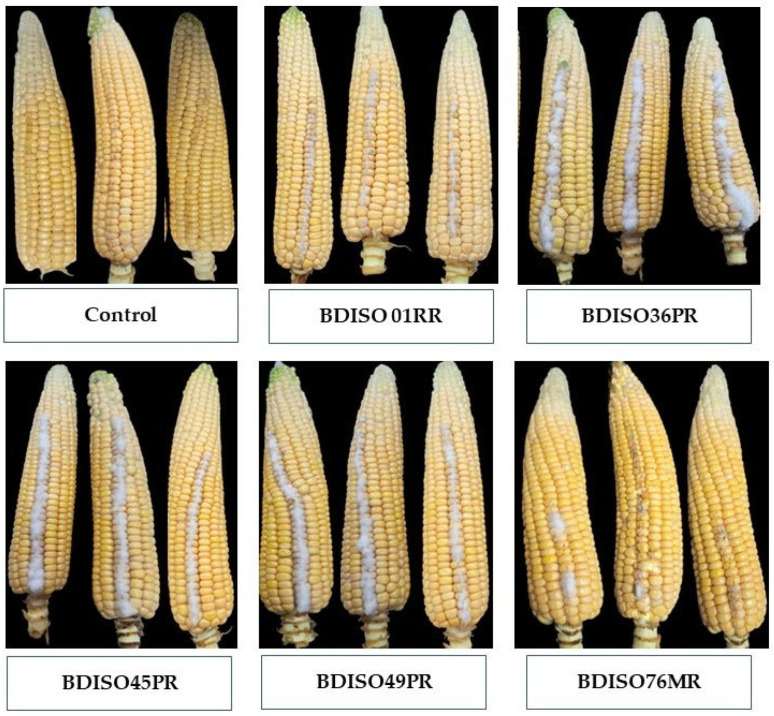
In vitro growth inhibition of *Fusarium proliferatum* on inoculated maize cobs by *Bacillus* spp. (BDISO01RR: *Bacillus amyloliquefaciens*, BDISO76MR: *Bacillus subtilis*, BDISO36PR: *Bacillus subtilis*, BDISO45PR: *Bacillus subtilis* and BDISO49PR: *Bacillus subtilis*). Untreated control: Sterile water. Photographs were taken 7 days after inoculation.

**Figure 2 toxins-16-00337-f002:**
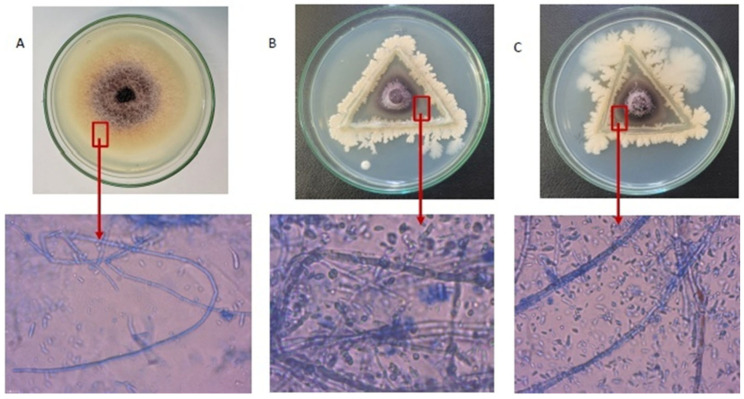
Morphological changes in *F. proliferatum* hyphae by *Bacillus* (*B. subtilis and B. amyloliquefacience*), observed under a compound microscope (40× magnification) at 7 DAI (**A**) Control, (**B**) BDISO01RR *(B. amyloliquefacience*), and (**C**) BDISO76MR (*B. subtilis*).

**Figure 3 toxins-16-00337-f003:**
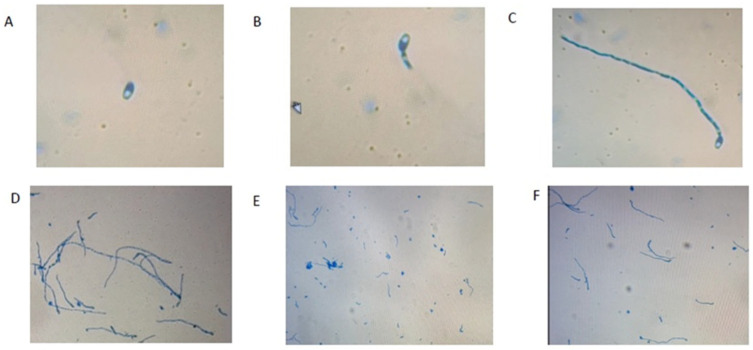
In vitro effect of *Bacillus* strains on conidial germination of *F. proliferatum* after 24 h incubation ((**E**) BDISO01RR (*B. amyloliquefaciense*) (**F**) BDISO76MR (*B. subtillis*)). The Conidia germination was assessed based on (**A**) non-germinated conidia (no hyphae), (**B**) germinated conidia with half hyphae, (**C**) germinated conidia with full hyphae, and (**D**) control (observed under a compound microscope (40× magnification)).

**Table 1 toxins-16-00337-t001:** Reduction in fumonisins accumulation in inoculated maize grain with different *Bacillus* strains and *F. proliferatum*.

Treatments	In Vitro	
	Fumonisin Concentration (ppm)	% Reduction over Control *
Untreated Control	245.57 ± 2.90 ^a^	0.00
BDISO36PR	40.10 ± 4.63 ^b^	83.67
BDISO01RR	5.06 ± 0.69 ^b^	97.94
BDISO49PR	9.55 ± 0.43 ^b^	96.11
BDISO45PR	33.10 ± 7.38 ^b^	86.52
BDISO76MR	3.63 ± 1.18 ^b^	98.52
Level of significance	***	
LSD	39.71	
%CV	0.23	

“***” represents significance at a 0.1% level of probability. Values with the same letter are statistically similar. Data are the averages of three replications. Each value represents the mean of three replications [BDISO76MR (*B. subtilis)*, BDISO01RR (*B. amyloliquefaciens*), BDISO36PR (*B. subtilis*), BDISO45PR (*B. subtilis*) and BDISO49PR (*B. subtilis*). Untreated control: seed treatment with only distilled water. Three replicates were used for each treatment. * %Reduction over control = [(Fumonisin concentration in uninoculated maize grains − Fumonisin concentration in inoculated maize grains)/Fumonisin concentration in uninoculated maize grains] × 100.

**Table 2 toxins-16-00337-t002:** Effects of *Bacillus* isolates on conidial formation and germination of *F. proliferatum* in vitro.

Treatments	Conidiation of *F. proliferatum* (×10^3^/mL)			% Reduction in Conidia Formation over Control *			% Germination of Conidia of *F. proliferatum* over Control		% Reduction in Conidia Germination over Control **	
	12 HPI	24 HPI	72 HPI	12 HPI	24 HPI	72 HPI	12 HPI	24 HPI	12 HPI	24 HPI
Untreated Control	23.00 ± 1.00 ^a^	47.00 ± 1.73 ^a^	140.67 ± 4.93 ^a^	0	0	0	69.00 ± 3.61 ^a^	87.84 ± 7.17 ^a^	0	0
BDISO36PR	16.00 ± 2.65 ^b^	32.00 ± 5.29 ^b^	97.00 ± 15.13 ^b^	30.43	31.91	31.04	59.70 ± 4.73 ^a^	80.52 ± 7.89 ^ab^	17.34	22.18
BDISO01RR	7.00 ± 1.73 ^de^	14.00 ± 2.65 ^de^	40.67 ± 10.02 ^d^	69.57	70.21	71.09	49.74 ± 7.02 ^a^	45.42 ± 4.64^de^	34.31	48.33
BDISO49PR	12.00 ± 0.50 ^c^	24.00 ± 1.00 ^c^	72.00 ± 3.00 ^b c^	47.83	48.94	48.82	50.10 ± 7.15 ^a^	59.49 ± 9.56 ^cd^	43.02	32.09
BDISO45PR	9.00 ± 1.00 ^d^	18.00 ± 2.65 ^d^	55.33 ± 7.57 ^cd^	60.87	61.70	60.66	54.10 ± 3.65 ^a^	68.46 ± 1.75 ^bc^	3.40	8.36
BDISO76MR	5.00 ± 1.00 ^e^	11.00 ± 1.73 ^e^	32.67 ± 4.93 ^d^	78.26	76.60	77.01	45.83 ± 1.89 ^a^	33.84 ± 3.42 ^e^	57.03	61.56
Level of significance	***	***	***	-	-	-	NS	***	-	-
LSD	2.65	5.08	27.121	-	-	-	45.27	15.55	-	-
%CV	0.12	0.12	21.12	-	-	-	0.48	0.14	-	-

“***” represents significance at a 0.1% level of probability. Values with the same letter are statistically similar. Data are the averages of three replications. Each value is the average of three tests for different types of *Bacillus*. Control: only *F. proliferatum.* * %Reduction in conidia formation = [(conidial formation in uninoculated maize grains −conidial formation in inoculated maize grains)/ conidial formation in uninoculated maize grains] × 100. ** %Reduction in conidia germination = [(conidial germination in uninoculated maize grains − conidial germination in inoculated maize grains)/conidial germination in uninoculated maize grains] × 100.

**Table 3 toxins-16-00337-t003:** Effects of different *Bacillus* strains on *F. proliferatum* in seed treatment and co-inoculated maize cob in the field.

Treatments	Seed Treatment		Co-Inoculation	
	Fumonisin Concentration (ppm)	%Reduction over Control *	Fumonisin Concentration (ppm)	%Reduction over Control *
Untreated Control	164.90 ± 13.06 ^a^	0.00	164.9 ± 13.06 ^a^	0.00
BDISO01RR	4.68 ± 0.75 ^b^	97.16	9.87 ± 1.60 ^b^	94.01
BDISO36PR	7.77 ± 1.25 ^b^	95.28	11.03 ± 2.70 ^b^	93.31
BDISO49PR	33.27 ± 10.48 ^b^	79.82	10.26 ± 2.40 ^b^	93.78
BDISO45PR	14.26 ± 1.10 ^b^	91.35	22.94 ± 6.35 ^b^	86.09
BDISO76MR	4.50 ± 1.32 ^b^	97.27	7.50 ± 2.50 ^b^	95.45
Level of significance	***		***	
LSD	20.00		17.83	
%CV	0.24		0.19	

“***” Represents significance at a 0.1% level of probability. Values with the same letter are statistically similar. Data are the averages of three replications. Control: seed treatment with only distilled water. * %Reduction over control = [(Fumonisin concentration in uninoculated maize grains − Fumonisin concentration in inoculated maize grains)/Fumonisin concentration in uninoculated maize grains] × 100.

## Data Availability

The corresponding author can provide the data from this study upon request.

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
