# Peer review of "Seed Priming with Rhizospheric Bacillus subtilis: A Smart Strategy for Reducing Fumonisin Contamination in Pre-Harvest Maize"

_toxins, 2024, doi:10.3390/toxins16080337_

Round 1
Reviewer 1 Report
Comments and Suggestions for Authors
Fumonisin contamination in maize is a significant risk worldwide. Biocontrol is an effective approach. This study tested five Bacillus strains in vitro condition and found all of them were effective against fumonisin production by colonizing the roots and inhibiting the mycelial growth of Fusarium proliferatum on maize grain. In field conditions, the BDISO76MR (B. subtilis) strains showed the highest efficacy for seed treatment and co-inoculation whereas seed treatment is better than co-inoculation. The above experiment was well-designed and obtained good results.
Howevere, there are some details that need to be revised:
1. The number in the chemical formula of inorganic substances should be lowercase in 5.1.3 and 5.1.4.
2. Three replications should be all represented in Figure 1.
3. SD value should be added in Table 1, 2 and 3.
4. The Conidiation of F. proliferatum (×103 /mL) at 12 and 24 HPI should also list the raw data in Table 2.
5. In L150, extra [] should be deleted.
6. Check the full manuscript, F. proliferatum should be in italics, and µl should be µL.
Author Response
Comment 1: The number in the chemical formula of inorganic substances should be lowercase in 5.1.3 and 5.1.4.
Response 1: Thank you for pointing out this. We have written the chemical formula according to the suggestion. The changes can be found on page no. 8, line no. 267; page no. 9, line no. 279-281
Comment 2: Three replications should be all represented in Figure 1.
Response 2: Thank you for your suggestion. We have added all the replications of the treatments in Figure 1, shown on page no. 3
Comment 3: SD value should be added in Table 1, 2 and 3.
Response 3: We appreciate your thoughtful observations. Please, find the revised manuscript where we added all the SD values in Table 1, 2, and 3. Page no. 3, 5, 6.
Comment 4: The Conidiation of F. proliferatum (×103 /mL) at 12 and 24 HPI should also list the raw data in Table 2.
Response 4: Thank you for your comment. The Conidiation of F. proliferatum (×103 /mL) at 12 and 24 HPI are listed in Table 2 (page no. 5.)
Comment 5: In L150, extra [] should be deleted.
Response 5: The extra [] in L150 is deleted which is now in L157 in the revised manuscript.
Comment 6: Check the full manuscript, F. proliferatum should be in italics, and µl should be µL.
Response 6: Thank you for pointing out this. In the revised manuscript we have written the F. proliferatum in italics and µL all over which is red marked throughout the manuscript.
Reviewer 2 Report
Comments and Suggestions for Authors
Field Potential of Native Bacillus in Controlling Fumonisin in Maize
Review Report: The manuscript is reviewed for consideration in toxins. The topic is of interest to food safety readers. Mycotoxins are serious concerns that affect not only the quality of food but also consumers' health. The methodology adopted is appropriate. However there are concerns which should be given consideration.
Title: the title should be revised, and should be the more fluent title of the manuscript
Line 34-42: Please add more information of the health hazards associated from FB1, FB2
The last paragraph should be revised showing previous researchs on the topic and then authors highlight the significant advancement in the field of their proposed title.
Limitation: Please incorporate the limitation and strengths of the manuscript
Author Response
Comment 1: The title should be revised, and should be the more fluent title of the manuscript.
Response 1: Thank you for your comment. We have revised our title and you can find it in the updated manuscript.
Comment 2: Line 34–42: Please add more information of the health hazards associated from FB1, FB2
The last paragraph has been revised to show previous research on the topic and then the authors highlight the significant advancements in their proposed title.
Response 2: Thank you for pointing out this. The revised manuscript now includes additional information about health hazards associated with FB1 and FB2, specifically in lines 35–36. Additionally, we incorporated previous work on Bacillus-based biocontrol for maize in our country into the revised last paragraph (lines 60–62).
Comment 3: Limitation: Please incorporate the limitation and strengths of the manuscript
Response 3: We agree with the suggestion. We have already mentioned our limitations in the discussion part, which you can find in lines 213, 214, 228–231. On the other hand, the strength of the manuscript is that we found a native Bacillus strain capable of controlling fumonisin contamination in pre-harvest maize.
Reviewer 3 Report
Comments and Suggestions for Authors
In the Materials and Methods section, subsection 5.3.2, Total fumonisin determination assay, the detection limit of the method must be specified.
Regarding the bibliographic references, even though there are many, almost half of them are older than 10 years. In my opinion, some of the older references should be replaced with more recent bibliographic references.
Comments on the Quality of English LanguageMinor editing of English language required
Author Response
Comment 1: In the Materials and Methods section, subsection 5.3.2, Total fumonisin determination assay, the detection limit of the method must be specified.
Response 1: Thank you for pointing out this. We have added the detection limit of the method. Please see the revised manuscript (Line: 389,390)
Comment 2: Regarding the bibliographic references, even though there are many, almost half of them are older than 10 years. In my opinion, some of the older references should be replaced with more recent bibliographic references.
Response 2: Thank you for your comment. We have replaced the older references in the revised manuscript.
Comment 3: Minor editing of English language required
Response 3: We appreciate your comment. We re-checked the whole manuscript and revised it accordingly. We corrected the grammar and other major English issues.
Round 2
Reviewer 3 Report
Comments and Suggestions for Authors
I have no other suggestions